# Vision-Language Models as Trainers for Instruction-Following Agents

## Abstract

Developing agents that can understand and follow language instructions is critical for effective and reliable human-AI collaboration. Recent approaches train these agents using reinforcement learning with infrequent environment rewards, placing a significant burden on environment designers to create language-conditioned reward functions. As environments and instructions grow in complexity, crafting such reward functions becomes increasingly impractical. To address this challenge, we introduce V-TIFA, a novel method that trains instruction-following agents by leveraging feedback from vision-language models (VLMs). The core idea of V-TIFA is to query VLMs to rate entire trajectories based on language instructions, using the resulting ratings to directly train the agent. Unlike prior VLM reward generation methods, V-TIFA does not require manually crafted task specifications, enabling agents to learn from a diverse set of natural language instructions. Extensive experiments in embodied environments demonstrate that V-TIFA outperforms existing reward generation methods under the same conditions.

## 1 Introduction

A central challenge in reinforcement learning (RL) research is developing agents that can reason abstractly, generalize across tasks, and communicate effectively. Language, whether natural or formal, plays a key role in enabling these abilities (Gopnik & Meltzoff, 1987). Recognizing this, many studies have explored incorporating language into RL to enhance communication, improve generalization and sample efficiency (Tellex et al., 2011; Mei et al., 2016; Goyal et al., 2019). The field can be broadly divided into language-conditioned RL (LC-RL), where language shapes the problem formulation (Anderson et al., 2018; Wang et al., 2019), and language-assisted RL, where language facilitates the agent's learning (Hu et al., 2019; Zhang et al., 2023). This work focuses on LC-RL, where the agent initially receives a language instruction and must act accordingly to follow that instruction. While RL provides a promising framework for training instruction-following agents, a major challenge is designing a reward function conditioned on language, which becomes increasingly difficult to implement efficiently as the complexity of the environment and language grows (Bahdanau et al., 2018). To scale instruction-following more broadly, an automated method is needed to evaluate whether the agent successfully completes the task specified by the instruction.

Prior work has explored replacing handcrafted language-conditioned rewards with methods that learn them indirectly from qualitative human inputs. A common approach is inverse RL (Ng et al., 2000), where the reward function is inferred from demonstrations paired with descriptions (Bahdanau et al., 2018; Fu et al., 2019). However, such high-quality language-annotated data can be elusive for complex and rare tasks. Meanwhile, for tasks without explicit language conditions, RL from human feedback (RLHF) has emerged as a powerful paradigm, allowing agents to learn from human guidance (Knox & Stone, 2009; Yuan et al., 2024). In RLHF, the reward function is learned by modeling human feedback, typically provided as comparative feedback (Christiano et al., 2017; Ibarz et al., 2018) or evaluative feedback (Wilde et al., 2021; White et al., 2024). This approach has shown promising results in enabling agents to perform low-level tasks like locomotion (Lee et al., 2021b) and manipulation (Hiranaka et al., 2023). However, RLHF for training instruction-following agents remains largely under-explored, likely because these tasks often involve multi-step, high-level reasoning, requiring humans not only to assess individual actions but also to account for long, compositional instructions. Consequently, gathering sufficient high-quality, language-annotated feedback for reward modeling in such settings is highly resource-intensive.

Both of these prevalent approaches to replacing manually handcrafted rewards rely heavily on human-provided data, limiting their scalability and generalizability. In response, the rise of foundation models (Radford et al., 2019; OpenAI, 2023; Reid et al., 2024) has sparked numerous efforts to reduce human supervision in designing reward functions by leveraging these models. One such approach involves generating code-based reward functions directly (Wang et al., 2024b; Xie et al., 2024; Ma et al., 2024). However, these methods often assume access to the environment's underlying code and low-level ground-truth states, making them difficult to scale to high-dimensional environments. Alternatively, pretrained vision-language models (VLMs), such as CLIP (Radford et al., 2021), have been employed to generate rewards by measuring the similarity between images and task descriptions in a shared vector space (Cui et al., 2022; Mahmoudieh et al., 2022; Rocamonde et al., 2024; Sontakke et al., 2024). Despite these advances, most approaches remain focused on single-objective tasks, often requiring manually crafted task specifications, such as demonstrations or text descriptions. In this paper, we aim to answer the question: *Can large vision-language models automatically generate rewards for training visual instruction-following agents, without relying on human data or direct access to the environment?*

To this end, we propose **V**ision-Language Models as **T**rainers for **I**nstruction-**F**ollowing **A**gents (V-TIFA), a method that leverages the advanced reasoning capabilities of large VLMs, such as Gemini (Reid et al., 2024), to automatically generate reward signals for training language-conditioned policies in the LC-RL setting. V-TIFA is inspired by the RLHF training paradigm, where the VLM acts as an evaluator, critiquing the agent's trajectories and delivering evaluative feedback (MacGlashan et al., 2017) to guide its learning. However, unlike conventional RLHF methods that require human annotators and explicit reward modeling (Christiano et al., 2017; White et al., 2024), V-TIFA directly uses feedback from the VLM to train the agent. This not only eliminates the need for costly human labor but also bypasses the reward modeling process, which can cause to reward misspecification and misgeneralization if not handle carefully (Casper et al., 2023). We evaluate V-TIFA in a set of challenging embodied environments from the ALFRED simulator (Shridhar et al., 2020), which includes 80 diverse human-generated language instructions. The results demonstrate that V-TIFA can be served as a proxy language-conditioned reward function, greatly outperforming prior VLM-based reward generation methods. Our key contributions are as follows:

- We introduce V-TIFA, a novel method that leverages VLMs to provide feedback for training instruction-following agents, eliminating the need for human-designed reward functions.
- With extensive experiments on a diverse set of instruction-following tasks, we show that V-TIFA can be served as an effective proxy for language-conditioned reward functions, consistently outperforms previous VLM-based reward methods.
- We conduct comprehensive analyses and ablation studies to explore the effectiveness of V-TIFA in training instruction-following agents, identifying the key factors contributing to its performance and robustness.

## 2 RELATED WORK

### 2.1 LANGUAGE-CONDITIONED RL

We position our work within the LC-RL framework (Luketina et al., 2019), where agent learns policies to complete tasks specified by instructions (MacMahon et al., 2006; Kollar et al., 2010; Wang et al., 2016). Prior works have explored this problem in the context of instruction-following, using RL to derive language-conditioned policies with environment rewards (Janner et al., 2018; Co-Reyes et al., 2018; Jiang et al., 2019; Chan et al., 2019). These approaches have been largely studied in either 2D spatial games (Bahdanau et al., 2018; Chen et al., 2019; Mirchandani et al., 2021) or 3D navigation and manipulation environments (Misra et al., 2014; MacGlashan et al., 2015; Hermann et al., 2017) with template instructions. By contrast, we focus on vision-language navigation (Anderson et al., 2018) using human-generated language instructions, without relying on environment rewards. We utilize ALFRED simulator (Shridhar et al., 2020), which offers diverse visually realistic household tasks with crowd-sourced language instructions. This challenging benchmark enables us to evaluate the recognition and reasoning capabilities of various VLMs in reward generation.

## 2.2 RL in the Absence of Reward Functions

Designing hard-coded reward functions in language-grounded environments often requires significant human effort. In CALVIN (Mees et al., 2022), for instance, rewards are computed by checking changes between initial and final states, relying on global state. In ALFRED (Shridhar et al., 2020), reward computation is even more complex, not only requiring the global state but also demonstrations to interpret instructions. To circumvent this, many works have focused on learning reward functions conditioned on language from human data. A common approach utilizes inverse RL (Ng et al., 2000; Ho & Ermon, 2016) to recover reward functions from demonstrations, which are then used to optimize policies via RL (Bahdanau et al., 2018; Fu et al., 2019; Mirchandani et al., 2021; Nair et al., 2022b). However, this approach relies on expert data, making it impractical for tasks that non-experts cannot easily perform (Brown et al., 2019; Zhang et al., 2021). To address this, we leverage VLMs as language-conditioned reward functions for training policies, eliminating the need for demonstrations. For single-objective tasks, a more practical way for humans to provide data is through feedback (Knox & Stone, 2009), where the agent is trained either directly from the feedback or indirectly by learning reward models that represent it (Yuan et al., 2024; Casper et al., 2023). In the robotics domain, the most common approaches to learning from feedback are preference-based RL (Christiano et al., 2017; Ibarz et al., 2018; Lee et al., 2021a;b) and rating-based RL (RbRL) (Knox & Stone, 2008; Wilde et al., 2021; White et al., 2024). Our training paradigm aligns with RbRL, where each trajectory is critiqued by an evaluator. However, instead of human evaluators, we leverage VLMs for this process. Additionally, we learn directly from feedback rather than modeling it, as in (MacGlashan et al., 2017; Arumugam et al., 2019)

## 2.3 Large Foundation Models as Reward Functions

(Kwon et al., 2023) and (Hu & Sadigh, 2023) introduce large language models (LLMs) to design reward functions in text-based games. Building on this, subsequent works have demonstrated that LLMs can directly generate Pythonic code for reward functions (Yu et al., 2023; Wang et al., 2024b; Xie et al., 2024; Ma et al., 2024). However, these methods typically assume access to the environment's source code or global state. Additionally, many robotic tasks are visual, requiring the use of VLMs instead. (Mahmoudieh et al., 2022) is the first to successfully use CLIP to train manipulation tasks based on language descriptions, but they require fine-tuning the CLIP on task-specific datasets. Recent works (Rocamonde et al., 2024; Sontakke et al., 2024) find that pretrained VLMs can potentially be used as reward functions without fine-tuning, by measuring the similarity between the images and text descriptions in the embedding space. However, these reward signals are often noisy and heavily dependent on task specifications (Rocamonde et al., 2024; Sontakke et al., 2024). Furthermore, these similarity-based reward functions lack explicit reasoning about tasks. (Wang et al., 2024a) is the first to use large VLMs to explicitly reason and provide preference labels for learning reward functions, which are then used to learn low-level control tasks. Most of these methods depend on manually crafted task descriptions and are limited to single-objective tasks, where the descriptions are often tailored to fit VLMs. Unlike these approaches, our method is robust to task descriptions, enabling multi-step, high-level reasoning from human-generated, compositional instructions, which allows for learning of language-conditioned policies. (Du et al., 2023) addresses a similar problem to ours, where they fine-tune a Flamingo VLM (Alayrac et al., 2022) on a carefully crafted dataset to detect task success. However, they do not train language-conditioned policies, leaving it unclear how robust their approach is under optimization pressure. By contrast, we show that VLMs without fine-tuning, equipped with simple prompting techniques, are effective for training agents directly.

## 3 Preliminary

**Language-conditioned RL.** We consider an augmented Partial Markov Decision Process (MDP) $\mathcal{M}$, defined by the tuple $(\mathcal{S}, \mathcal{O}, \mathcal{A}, P, R, \mathcal{L}, \gamma)$, where $\mathcal{S}$ is the state space, $\mathcal{O}$ is the observation space, $\mathcal{A}$ is the action space consisting of primitive actions—in ALFRED, these include navigation and interaction actions (MoveAhead, Pickup, ToggleOn, etc.), $P(s'|s,a)$ is the transition probability, $\gamma \in [0,1]$ is the discount factor, $\mathcal{L}$ is the space of language instructions from which the task instruction $l$ is drawn, and $R : \mathcal{S} \times \mathcal{A} \times \mathcal{S} \times \mathcal{L} \to \mathbb{R}$ is a language-conditioned state action reward function. The agent takes actions based on a language-conditioned policy

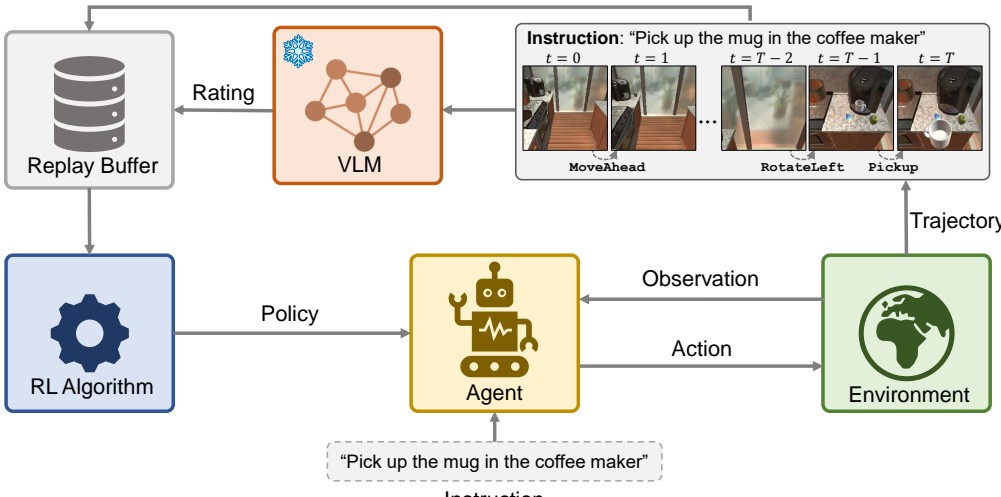

Figure 1: **V-TIFA Overview**: A pretrained VLM acts as an evaluator, delivering ratings based on the observed agent actions and state transitions. These ratings serve as reward signals for training the language-conditioned policy using any off-policy RL algorithm.

$\pi(a|s,l) : \mathcal{S} \times \mathcal{L} \rightarrow \mathcal{A}$. The goal of RL algorithms is to learn a policy that maximizes the expected return $\mathbb{E}_{\pi,l\sim\mathcal{L}}[\sum_{t=0}^{T-1} \gamma^t R(s_t, a_t, s_{t+1}, l)]$, where $T$ is the trajectory horizon.

Let $\tau = (o_t, a_t)_{t=0}^{T-1} = (o_0, a_0, \ldots, o_{T-1}, a_{T-1}, o_T)$ denote a trajectory composed of a sequence of observations and actions. In this work, we consider sparse reward problems, where the agent is rewarded at the end of the trajectory, indicating whether the agent successfully completes the instruction $l$. In ALFRED, the ground-truth reward function returns 1 when the agent completes the instruction and 0 otherwise. Additionally, the trajectory terminates either when the instruction is completed or when a timeout occurs, resulting in varying trajectory lengths.

**Rating-based RL.** When the reward function $R$ is unavailable, standard RL algorithms cannot be used to derive policies. Instead, we assume that an annotator critiques the trajectory $\tau$, along with the task instruction, by assigning a rating $c$ from the set $\mathcal{C} = \{0, 1, \ldots, n-1\}$, where 0 is the lowest possible rating and $n-1$ is the highest, indicating the quality of the trajectory. Descriptive labels can also be assigned to the rating levels. For example, with $n = 4$ rating levels, level 0 could be labeled "very bad", level 1 "bad", level 2 "good", and level 3 "very good". Unlike previous work (Wilde et al., 2021; White et al., 2024), which focuses on learning an explicit human-aligned reward function, we directly use feedback from the annotator (in our case, vision-language models) to train the policy, following a similar approach to (MacGlashan et al., 2017; Arumugam et al., 2019).

**Vision-language models.** In this paper, we define vision-language models (VLMs; (Zhang et al., 2024)) as models capable of processing both language inputs $p = (x_0, \ldots, x_m)$, where $x_m \in \mathcal{V}$, and a visual input $I \in \mathcal{I}$. Here, $\mathcal{V}$ represents a finite vocabulary, and $\mathcal{I}$ denotes the space of RGB images. Given these inputs, the VLM $H$ generates language outputs as $y = H(p, I)$, where $y = (y_0, \ldots, y_k)$ and $y_k \in \mathcal{V}$. We focus on VLMs trained on diverse text and image datasets, which enables them to generalize effectively across different environments and task instructions. Moreover, these models must be capable of answering questions based on a single image (OpenAI, 2023; Anthropic, 2024; Reid et al., 2024), a crucial ability for accurately rating trajectories.

## 4 METHOD

**Overview.** V-TIFA leverages the advanced reasoning abilities of pretrained VLMs to deliver feedback for training instruction-following agents through online RL. This is achieved by assigning a rating at the end of the trajectory, reflecting how likely the agent successfully completed the given instruction. Unlike prior rating-based RL methods that require human involvement during training, our method fully automates the generation of evaluative feedback, allowing agents to train without

---

**Algorithm 1** V-TIFA training algorithm.

---

1: **Input**: A pretrained VLM $H$, visual prompt constructor $\Omega$, textual prompt constructors for summarizing $\Psi_S$ and rating $\Psi_R$
2: **Initialize**: Policy $\pi_\theta$, replay buffer $\mathcal{R}$.
3: **while** not converged **do**
4:     Sample instruction $l_i \sim \mathcal{L}$
5:     Run $\pi_\theta$ to collect trajectories $\{\tau_i\}$ given $l_i$
6:     **for each** $\tau_i$ **do**
7:         Construct prompts for summarization: $I_S = \Omega(\{o_t\}_{t=0}^T)$ and $p_S = \Psi_S(\{a_t\}_{t=0}^{T-1}, l_i)$
8:         Query for summarization: $S = H(p_S, I_S)$
9:         Construct prompt for rating: $p_R = \Psi_R(S, l_i)$
10:        Query for rating: $c_i = H(p_R, l_i)$
11:     **end for**
12:     Store trajectories into replay buffer: $\mathcal{R} \leftarrow \mathcal{R} \cup \{(l_i, \tau_i, c_i)\}$
13:     Optimize policy $\pi_\theta$ using data sampled from $\mathcal{R}$ with any off-policy RL algorithm
14: **end while**

---

human intervention. An overview of V-TIFA is shown in Figure 1, and the detailed training procedure is provided in Algorithm 1. The agent first receives a language instruction $l_i$, then interacts with the environment to collect trajectories $\{\tau_i\}$ based on the policy $\pi_\theta$. Each trajectory $\tau_i$, along with the instruction $l_i$, is sent to the VLM to obtain a corresponding rating $c_i$. These trajectories, along with the corresponding instructions and ratings $\{(l_i, \tau_i, c_i)\}$, are then stored in the replay buffer $\mathcal{R}$. Finally, the RL algorithm updates the policy $\pi_\theta$ using data sampled from the replay buffer.

Prior work in RbRL (Yuan et al., 2024; White et al., 2024) typically requires a reward modeling step, as directly using human feedback is prohibitively expensive for RL systems. However, learning a reward model conditioned on language introduces further complexity, as it must account for multiple tasks. This requires a large amount of instruction-dependent trajectories to develop a reward function that generalizes effectively (Nair et al., 2022a; Karamcheti et al., 2023; Ma et al., 2023). By contrast, we incorporate VLMs directly into the training loop, eliminating the reward modeling step—a process that, if not carefully managed, can be prone to reward misspecification and misgeneralization (Casper et al., 2023).

**VLMs for Rating.** In the LC-RL problem, language instructions can be complex and highly compositional. For instance, an instruction like "*Put the coffee cup in the sink, turn on the water, turn off the water and pick up the coffee cup*" involves multiple sub-tasks. As a result, an automatic evaluator should be fine-grained enough to evaluate trajectories accurately based on the specific language instruction. Moreover, multiple successful policies can produce diverse yet valid trajectories for the same instruction. Evaluating these solely on final outcomes can be misleading, especially with highly compositional instructions, where critical sub-tasks may be completed at different stages within the trajectory. To ensure that VLMs provide accurate ratings, we prompt the model with the entire trajectory, which includes visual observations, actions, and the corresponding instruction. Figure 2 illustrates this prompting process. First, we query the VLM to generate a free-form summary of the trajectory. This summary is then used to prompt the VLM for a final rating. Since the VLM processes individual images, querying it for each visual observation can be inefficient and may limit its ability to capture temporal dynamics. To address this, we use a combination of visual and textual prompts to efficiently represent the full trajectory. Our approach to visual prompting is inspired by recent work (Jia et al., 2022; Bar et al., 2022; Shtedritski et al., 2023), which shows that pretrained VLMs can enhance visual reasoning capabilities.

Concretely, let $\Omega$ be the visual prompt constructor, and $\Psi_S$ and $\Psi_R$ be the textual prompt constructors for summarization and rating, respectively. Given a trajectory, $\Omega$ maps the visual observations into a new image, $I_S = \Omega(\{o_t\})$, by concatenating the image observations and placing a timestep caption under each individual image. $\Psi_S$ maps the actions and instruction $l$ into a text prompt, $p_S = \Psi_S(\{a_t\}, l)$. This prompt contains information about the trajectory's length, executed actions, and a question to evaluate the completion of the instruction $l$. The summary of the trajectory is then obtained from the VLM as $S = H(p_S, I_S)$. For the rating, we construct a prompt using the generated summary and instruction $l$ as $p_R = \Psi_R(S, l)$, and then query the VLM for the final rating $c = H(p_R, l)$. In $\Psi_R$, we specify the rating range and assign descriptive labels for the lowest and highest ratings. Figure 2 illustrates $\Omega_S$ and $\Psi_S$ in the yellow box, and $\Psi_R$ in the blue box.

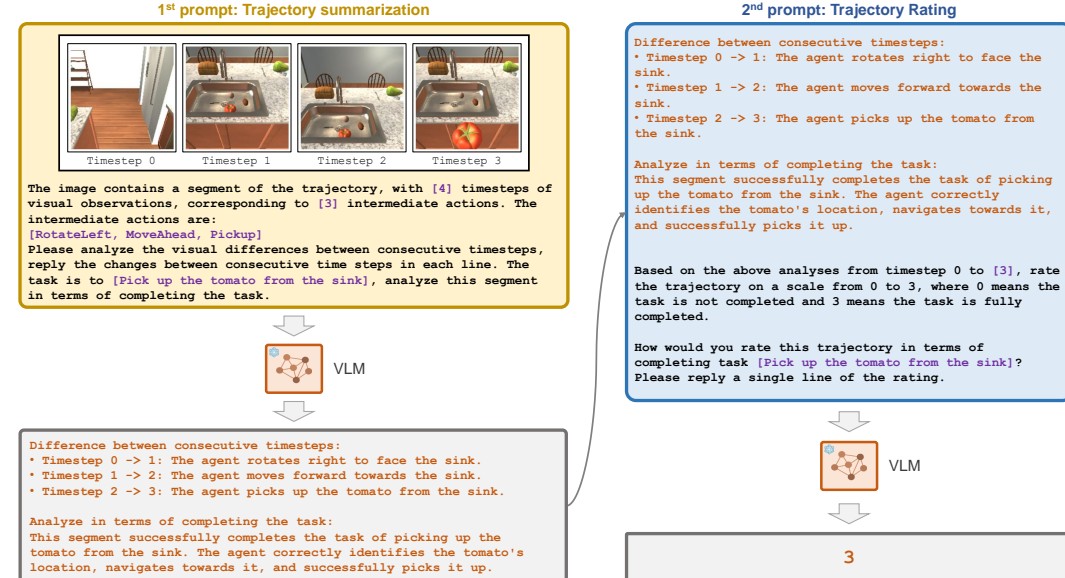

Figure 2: Given an instruction and a trajectory collected by the agent, we construct visual and textual prompts to query the VLM for a summary of the trajectory and an evaluation of how well it completes the instruction (yellow box). The summary is then used to construct a prompt to query for a final rating from the VLM (blue box). An example of the summary and rating is shown in the gray box. The template shown here is applied across all instructions and environments in the paper.

While (Cabi et al., 2020) also explores per-frame annotation with human involvement, our approach leverages VLMs to automate the annotation process, eliminating the need for human intervention.

**Implementation Details.** The trajectories can vary in length, reaching up to 50 steps in our environments. Concatenating a large number of images may increases inference time and degrade the reasoning performance of VLMs, as their limited input size necessitates downscaling when the input exceeds the model's capacity. In practice, we divide each trajectory into segments (*e.g.*, 10 steps per segment) during summarization. These segment summaries are then concatenated to form the final summary. For trajectory rating, since the input is purely text, large language models could be used. However, for simplicity, we use the same VLM for both summarization and rating.

## 5 EXPERIMENTAL EVALUATION

The goal of our experiments is to evaluate V-TIFA's effectiveness in training instruction-following agents in the online RL setting. We compare V-TIFA to prior VLM-based reward generation methods in visual household tasks from the ALFRED simulator (Shridhar et al., 2020). While previous work has primarily focused on low-level control tasks, we extend these methods to this challenging benchmark. Concretely, we aim to answer the following questions:

1. *How does the effectiveness of V-TIFA compare to other methods in LC-RL setting?*
2. *What aspects of V-TIFA are crucial for its success?*
3. *How consistent and effective is the feedback quality across pretrained VLM models?*
4. *What advantages does evaluative feedback have over comparative feedback?*

### 5.1 EXPERIMENTAL SETTINGS

**ALFRED Environment.** We evaluate methods in a set of challenging embodied environments (Figure 3), including Kitchen, Bathroom, Living Room, and Bedroom, drawn from the valid-unseen folds of the ALFRED simulator (Shridhar et al., 2020). Unlike other synthetic LC-RL benchmarks that rely on template instructions (Hermann et al., 2017; Chevalier-Boisvert et al., 2019), ALFRED offers visually realistic environments with crowd-sourced language instructions. This allows us to evaluate the VLMs' ability to generate effective rewards across complex, natural language directives. We leverage a modified version of the ALFRED simulator (Zhang et al., 2023), which allows for on-

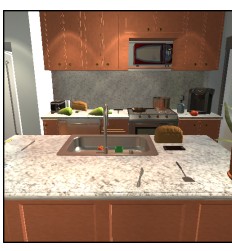 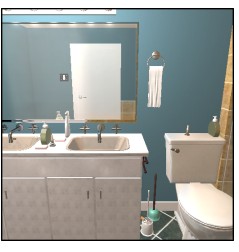 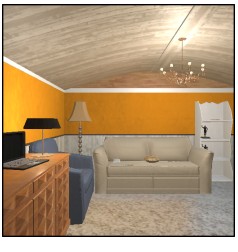 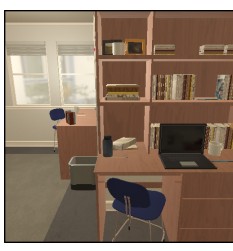

(a) Kitchen      (b) Bathroom      (c) Livingroom      (d) Bedroom

Figure 3: We evaluate V-TIFA in four embodied environments from the ALFRED simulator, where the goal is to train an agent to follow natural language instructions to complete household tasks.

line RL interaction via a gym interface. ALFRED abstracts away low-level control into 12 discrete actions (*e.g.*, MoveAhead, Pickup), along with 82 discrete object types. In these environments, agents are randomly situated in rooms and perceive the environment through proprioceptive information and $224 \times 224$ egocentric RGB image observations. In each environment, we define 10 evaluation tasks, each composed of 2 sub-tasks with their own instruction, resulting in a total of 80 instructions across the four environments. The agent is considered successful if both sub-tasks are completed. Further details about the environments and instructions are provided in the Appendix.

**Baselines.** We compare V-TIFA to prior baselines that also leverage pretrained VLMs (without fine-tuning) to generate rewards based on the task description and the agent's visual observations. These baselines involve contrasting the text embedding with either a single image embedding or multiple image embeddings:

- GT Reward: This baseline trains the agent using ground-truth rewards from the environment and serves as an upper bound. At the end of the trajectory, the agent receives a reward of 1 for completing the instruction and 0 otherwise.
- CLIP Reward: The reward is generated by computing the cosine similarity between the final observation and the language instruction in CLIP embedding space (Radford et al., 2019). This reward computation method has also been explored in (Cui et al., 2022; Mahmoudieh et al., 2022; Rocamonde et al., 2024).
- R3M Reward: This method was originally designed for representation learning in robotics (Nair et al., 2022b). We leverage the pretrained predictor from R3M, which takes the initial and final observation along with the language instruction to output a score measuring how well the instruction aligns with the temporal dynamics between the two images. This reward computation is investigated in (Adeniji et al., 2023).
- RoboCLIP Reward: Similar to the CLIP Reward, the reward is generated by computing the similarity between video observations and a demonstration video in S3D embedding space (Xie et al., 2018). However, since our method does not assume access to task demonstrations, we instead use the text-based version of RoboCLIP (Sontakke et al., 2024).

For all baselines, the generated rewards are obtained at the end of each trajectory, and we use default task instructions from ALFRED as the task description for computing rewards.

**Training and Evaluation Procedure.** In our experiments and baselines, we use a variant of Implicit Q-Learning (IQL) (Kostrikov et al., 2022) as the off-policy RL algorithm to train the policy, as it has been shown to successfully train agents in the ALFRED (Zhang et al., 2023). We train agents for $800k$ steps in Bedroom, and $500k$ steps in the others. Success rates are measured every 100 epochs, averaged over 500 episodes. Note that success is defined as 1 when the agent successfully completes both sub-tasks. For V-TIFA, we use Gemini-1.5-Pro (Reid et al., 2024) as the pretrained VLM, with 4 levels of ratings, and divide the trajectory into segments of 10 steps. We perform experiments on a PC with an AMD Ryzen 7906X and two RTX 4090 GPUs, with a training time for V-TIFA of approximately 1.5 days per run. Further details are provided in the Appendix.

### 5.2 EFFECTIVENESS OF V-TIFA FOR TRAINING INSTRUCTION-FOLLOWING AGENTS?

We first examine whether V-TIFA can provide reward signals for learning language-conditioned policies. Figure 4 shows the success rate over the course of training across three runs. The results show that V-TIFA consistently outperforms other baselines across environments, coming closest to

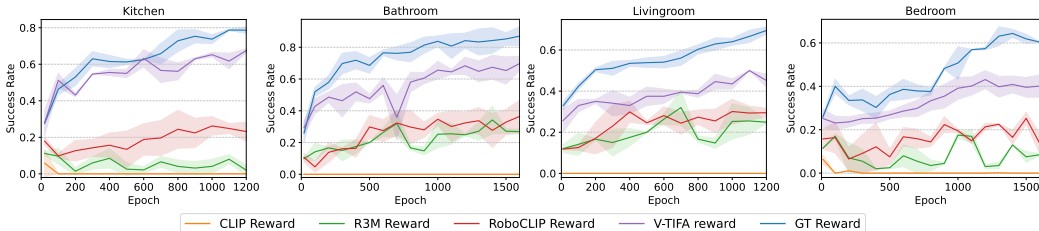

Figure 4: Success rate over training course of all methods in four environment. V-TIFA greatly outperforms all baselines across environments, and closest to GT Reward in Kitchen and Bathroom. The solid line is the mean success rate, while the shaded regions is to the standard deviation, both calculated across three different random seeds.

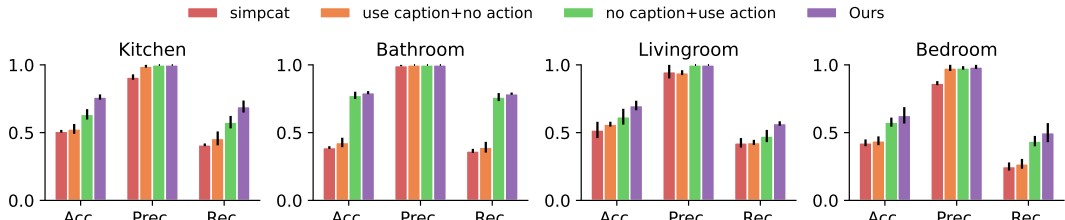

Figure 5: Effect of different components in our trajectory summary prompt. Overall, including actions in the summary prompt has the most significant impact.

GT Reward in the Kitchen and Bathroom environments. Among the baselines, we find that the CLIP Reward fails to guide agent learning in solving tasks. This is likely because CLIP is pretrained on single images, and its similarity score lacks the temporal understanding required to capture the sequential nature of instructions. Our findings are consistent with (Sontakke et al., 2024), which similarly highlights CLIP's limitations in handling temporal dynamics. On the other hand, both R3M Reward and RoboCLIP Reward provide some useful signals for policy learning, with RoboCLIP performing better in 2 out of 4 environments. This is because RoboCLIP uses pretrained video-language models, which capture richer temporal dynamics than R3M, which is only pretrained to align language with the initial and future frames. In contrast, V-TIFA performs explicit reasoning over the trajectory and accounts for action-driven changes in transitions, resulting in more accurate reward signals grounded in the agent's behavior, which leads to significantly improved performance.

## 5.3 ANALYZING V-TIFA

While V-TIFA successfully provides reward signals for policy training, a visible gap remains between V-TIFA and the GT Reward. In this section, we examine the underlying reasons for this gap and analyze the impact of various design decisions in V-TIFA. Additionally, we evaluate the effectiveness of various large pretrained VLMs. To perform these experiments, we collect trajectories from GT Reward agents along the training course. In each environment, for each checkpoint, we record 40 trajectories corresponding to 2 trajectories per instruction, resulting in approximately 500 trajectories per environment, with the averaged return over collected dataset about 0.6-0.7. We then use the same prompt to query the VLM for ratings. To enable direct comparison with ground-truth rewards, we assign a reward of 1 to ratings at the maximum value, and 0 otherwise. The intuition is that when a trajectory successfully completes an instruction, the rating should be at its highest. We then measure accuracy (Acc.), precision (Prec.), and recall (Rec.) to evaluate the performance.

**Alignment of V-TIFA with Ground-Truth Rewards.** As shown in Figure 5, we observe that the accuracy of V-TIFA is highest in Kitchen and Bathroom, at roughly 80%, while it reaches 65% in the other environments. These results are also reflected in the final performance of the trained agents in Figure 4, where V-TIFA comes closer to the GT Reward agent's performance in the Kitchen and Bathroom. Interestingly, the precision of V-TIFA remains close to 1 across all environments, suggesting that VLMs rarely assign the maximum rating to failed trajectories. In other words, when VLMs give the highest rating, the trajectory has almost always successfully completed the instruction. However, the recall of V-TIFA is somewhat lower than its accuracy and precision, as the VLM often assigns a rating of 2 rather than the maximum rating of 3 to several successful trajectories.

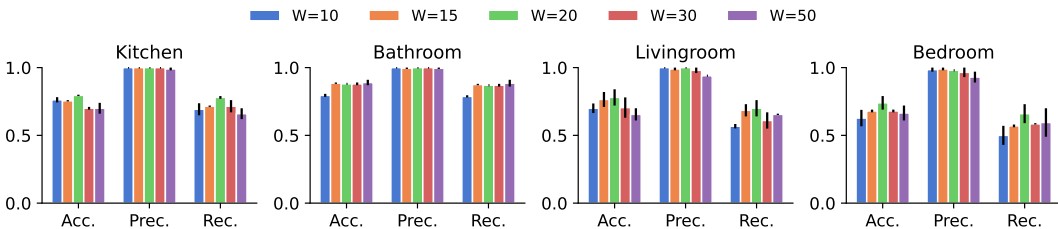

Figure 6: Effect of different segment lengths on performance. The performance varies only slightly across different segment lengths.

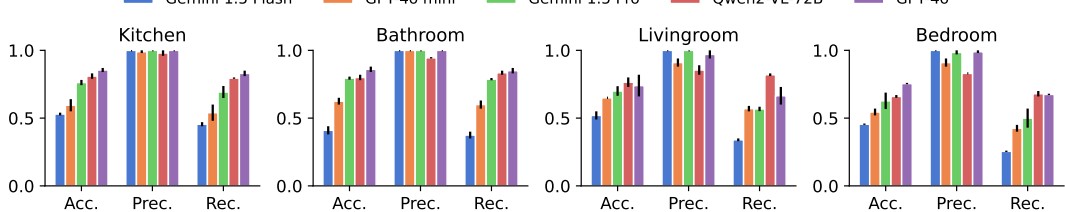

Figure 7: We investigate the performance of different large pretrained VLMs. Bigger models achieve better performance but at the cost of increased inference time.

**Effectiveness of Components in Summary Prompt.** We ablate several design decisions as follows: (1) simply concatenate image observations and exclude actions from the prompt (*simpcat*), (2) include a timestep caption under each image but exclude actions (*use caption+no action*), and (3) include actions in the prompts but exclude captions (*no caption+use action*). Figure 5 illustrates the performance of different prompt configurations. The results clearly show that including actions in the summary prompt contributes to the greatest improvement, while adding captions offers a slight advantage over simple image concatenation.

**Effectiveness of segment size.** Figure 6 shows the effect of different segment lengths on performance. The results indicate that performance varies only slightly across lengths. Although a segment length of 20 achieves the best results, it comes with increased inference time due to larger images. To balance effectiveness and efficiency, we use a segment length of 10 in our experiments.

**Effectiveness of VLMs.** We further investigate the effectiveness of different large pretrained VLMs, including Gemini 1.5 Flash, Gemini 1.5 Pro (Reid et al., 2024), GPT-4o Mini (OpenAI, 2024b), GPT-4o (OpenAI, 2024a), and Qwen2-VL (Bai et al., 2023). For Qwen2-VL, we use the released model from the authors and run it on a single A100 GPU. Figure 7 and Table 1 present the performance and inference time for the querying process of the large VLMs. Lite models, such as Gemini 1.5 Flash and GPT-4o Mini, often exhibit poorer performance. Although GPT-4o achieves the best performance among the models considered, we find that GPT-4o models are unstable during training, occasionally returning null text. Additionally, their inference times are inconsistent (e.g., GPT-4o Mini is slower than GPT-4o) and generally slower than Gemini 1.5 Pro. While Qwen2-VL shows promising results with the second-best performance, its inference time on images is significantly slower due to limited resources. Therefore, we select Gemini 1.5 Pro for more efficient training in our experiments.

## 6 DIFFERENT FEEDBACK TYPES

In this section, we explore a different type of feedback: comparative feedback. In this setup, we query the VLM for comparative feedback on the collected trajectories from Section 5.3, denoted as $D_{target}$. Since this feedback type requires the evaluator to compare pairs of trajectories, we additionally collect an extra dataset of the double size for comparison, denoted as $D_{reference}$. We simulate the querying process as follows: for each trajectory in $D_{target}$, we uniformly sample a trajectory from $D_{reference}$ with the same task instruction. We then use the same summary prompt to summarize both trajectories and utilize the comparison prompt from (Wang et al., 2024a) to obtain the preference. Specifically, if the VLM prefers the trajectory from $D_{target}$, we assign a reward of 1 to that trajectory, and 0 otherwise. We use the same evaluation metric as in Section 5.3.

Table 1: The inference time of the querying process at each step for different VLMs

| Model | Summary (s) | Rating (s) | Total time (s) |
|---|---|---|---|
| Gemini 1.5 Flash | 15.7 | 1.3 | 17 |
| GPT-4o mini | 38.8 | 0.8 | 39.6 |
| Gemini 1.5 Pro | 25 | 1.6 | 26.6 |
| Qwen2-VL-72B | 912.2 | 3.6 | 915.8 |
| GPT-4o | 29.5 | 0.8 | 30.3 |

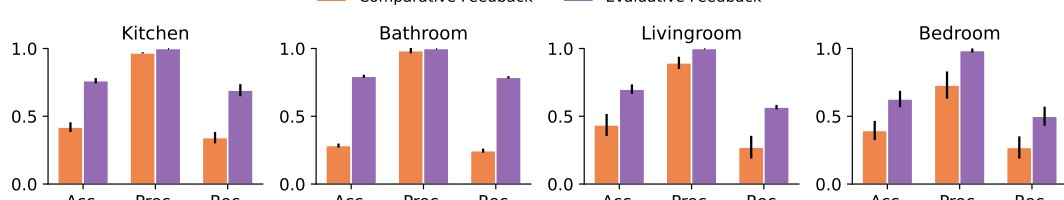

Figure 8: The comparison between comparative feedback and evaluative feedback.

The results in Figure 8 indicate that comparative feedback leads to poorer performance. To better understand the underlying cause, we manually inspect the summaries and responses from the VLM. Our analysis reveals that the VLM frequently favors shorter trajectories, even when both successfully complete the instruction. For example, for the task "*Pick up the tomato from the sink*", the agent's random initial position can result in varying distances from the target object, making longer trajectories not necessarily worse than shorter ones. Because of its binary nature, this type of feedback does not convey the degree to which one sample is better or worse than another. This limitation of comparative feedback has also been noted in (Casper et al., 2023; Wang et al., 2024a; White et al., 2024). To address this issue, previous works often require the collection of a large number of samples and the development of strategies to select informative reference samples (Biyik & Sadigh, 2018; Bıyık et al., 2020; Sadigh et al., 2017), which can be even more challenging in the LC-RL.

## 7 CONCLUSION

We present a method that leverages large vision-language models (VLMs) as a proxy for language-conditioned reward functions to train instruction-following agents. Our proposed prompt technique enables VLMs to explicitly evaluate the entire agent trajectory, providing a deeper understanding of the language instruction and generating more effective reward signals for training. Our experiments demonstrate that V-TIFA is robust to language instructions and consistently outperforms prior baselines across various embodied environments.

**Limitations.**     While V-TIFA successfully trains instruction-following agents in a language-conditioned reinforcement learning setting using vision-language models (VLMs) without fine-tuning, a noticeable gap remains compared to agents trained with ground-truth rewards. This discrepancy primarily arises from occasional inaccuracies in VLM feedback. Additionally, in the environments we tested, only large-scale VLMs delivered strong performance, though at the cost of increased inference time (approximately $2.5\times$ longer than training with environment rewards alone). Smaller models, while faster, yielded only moderate results. Future work could explore integrating advanced techniques such as self-correction (Miao et al., 2024) to improve the feedback consistency and accuracy of smaller VLMs. This would pave the way for more efficient, scalable reinforcement learning systems that maintain high performance while reducing computational overhead, making RL more feasible for deployment in real-world environments.

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

# A  APPENDIX

## A.1  RL ALGORITHM DETAILS

We utilize Implicit Q-Learning (IQL) (Kostrikov et al., 2022) with a transformer-based architecture for both the policy and critic networks, similar to (Zhang et al., 2023). The hyperparameters are provided in Table 2. The main difference is that we set the quantile parameter $\tau = 0.5$, making IQL a standard off-policy online RL algorithm, rather than one suited for offline RL as in (Zhang et al., 2023). During training, to reduce exploration time—which is particularly challenging in ALFRED—we seed the buffer with 2-3 human-collected demonstrations. It is important to note that while these demonstrations complete the task, they are not necessarily optimal. We use them to reduce exploration time; however, our method can without them, albeit with longer training times. This approach is applied consistently across all baselines. Additionally, we relabel the rewards in the seed buffer to align with each baseline's framework, ensuring compatibility during training.

| Parameter | Value |
|---|---|
| Batch Size | 128 |
| # Training Steps | 800k Bedroom, 500k otherwise |
| Learning Rate | $1e - 4$ |
| Optimizer | AdamW |
| Dropout Rate | 0.1 |
| Weight Decay | 0.1 |
| Discount $\gamma$ | 0.97 |
| Q Update Polyak Averaging Coefficient | 0.005 |
| Policy and Q Update Period | 8 per train iter |
| IQL Advantage Clipping | [0, 100] |
| IQL Advantage Inverse Temperature $\beta$ | 5 |
| IQL Qunatile $\tau$ | **0.5** |
| Maximum Context Length | 8 |

Table 2: Hyperparameters for IQL

## A.2  RL ENVIRONMENT DETAILS

The ALFRED benchmark (Shridhar et al., 2020) is originally designed for imitation learning. We use a modified version of ALFRED from (Zhang et al., 2023), which supports policy learning using reinforcement learning (RL). Also, we make further modifications to the environment to ensure that when objects are picked up, they remain clearly visible within the agent's view. In the original setup, the agent's view is often occluded by larger objects. This change allows VLMs to recognize objects more effectively. We define evaluation tasks by randomly sampling 10 tasks for each of the 4 unseen ALFRED floor plans, resulting in a total of 40 tasks. Each task is constrained to consist of 2 sub-tasks. For tasks with more than 2 sub-tasks, we only use the first 2. This is because, with longer tasks, the baseline RL algorithm from (Zhang et al., 2023) may fail to learn any tasks. All generated tasks from the floor plans are shown in Tables 3, 6, 5, and 4. The agent is considered successful if it completes both sub-tasks. Note that during training, the agent must complete the first sub-task before switching to the next. For the ground-truth reward function, the agent receives a reward of 1 whenever it completes a sub-task, then switches to the next sub-task or stops if the second sub-task is already completed. The observations provided to the agents are $224 \times 224$ RGB images. For all baselines, we first preprocess these images by passing them through a frozen ResNet-18 encoder (He et al., 2016) pretrained on ImageNet, resulting in $512 \times 7 \times 7$ observations. The action space of ALFRED consists of 5 navigation actions: MoveAhead, RotateRight, RotateLeft, LookUp, and LookDown, and 7 interaction actions: Put, Pickup, Open, Close, ToggleOn, ToggleOff, and Slice. For interaction actions, the policy also outputs one of 82 object types to interact with. Note that for the VLM summary prompt, we use only actions and not object types. Due to large discrete action space (5 + 7 * 82), we perform same masking as (Zhang et al., 2023) to prevent agents from taking actions that are not possible (*e.g.*, the policy cannot output Close for object Tomato).

| Task No. | Sub-task Type | Instruction |
|---|---|---|
| 1 | PickupObject | Pick up the spoon from the counter |
| | PutObject | Put the spoon in the white cup on the shelf. |
| 2 | PickupObject | Pick up the egg that is beside the fork in the sink. |
| | CoolObject | Open the refrigerator, then place the egg on the glass shelf and close the fridge. Wait then open the fridge and pick up the egg, then close the fridge. |
| 3 | PickupObject | Pick up the tomato from the sink. |
| | CoolObject | Open the fridge door, put the tomato inside of the fridge, close the door, open the door, take the tomato out, close the door. |
| 4 | PickupObject | Pick up the mug in the coffee maker |
| | CoolObject | Open the fridge, put the cup in the fridge, close the fridge, wait, open the fridge, pick the cup, close the fridge |
| 5 | PickupObject | Pick up the bread. |
| | CoolObject | Open the fridge, put the bread in the fridge, close the fridge, open the fridge, get the bread, and close the fridge. |
| 6 | PickupObject | Pick up the white coffee cup to the right of the trophy. |
| | CleanObject | Put the coffee cup in the sink, turn on the water, turn off the water and pick up the coffee cup. |
| 7 | PickupObject | Pick up the smaller silver knife on the counter. |
| | PutObject | Put the knife in the green cup in the sink. |
| 8 | PickupObject | Pick up a bowl from the shelf |
| | PutObject | Put the bowl on the counter |
| 9 | PickupObject | Grab the knife from the counter |
| | PutObject | Put the knife in the pan on the stove |
| 10 | PickupObject | Pick up the knife from the counter. |
| | CleanObject | Place the knife in the sink and turn the water on. Turn the water off and pick up the knife. |

Table 3: Tasks from Kitchen environment.

| Task No. | Sub-task Type | Instruction |
|---|---|---|
| 1 | PickupObject | Pick up the bowl from the shelf |
| | ToggleObject | Turn on the lamp sitting on the desk while holding the bowl |
| 2 | PickupObject | Pick up the white mug from the desk. |
| | ToggleObject | Turn the desk lamp on with the mug in hand. |
| 3 | PickupObject | Pick up the book from the bed. |
| | ToggleObject | Turn on the lamp on the desk while carrying the book |
| 4 | PickupObject | Pick up the mug from the shelf. |
| | ToggleObject | Turn the lamp on while holding the cup. |
| 5 | PickupObject | Pick up the bowl on the desk. |
| | ToggleObject | Turn on the lamp on the desk while holding the bowl. |
| 6 | PickupObject | Pick up the pencil from the desk |
| | PutObject | Put the pencil in the bowl |
| 7 | PickupObject | Pick up the alarm clock from the desk |
| | ToggleObject | Turn on the lamp on the desk while holding the alarm clock. |
| 8 | PickupObject | Pick up the clock from the back of the desk. |
| | ToggleObject | Hold the clock and turn on the lamp on the right side of the desk. |
| 9 | PickupObject | Pick up the mug on the shelf. |
| | PutObject | Put the mug on the desk. |
| 10 | PickupObject | Pick up the pencil on the desk. |
| | PutObject | Place the pencil in the glass bowl on the desk. |

Table 4: Tasks from Bedroom environment.

| Task No. | Sub-task Type | Instruction |
|---|---|---|
| 1 | PickupObject | Pick up the cell phone from the dresser |
|   | ToggleObject | Hold the cell phone and turn the lamp on |
| 2 | PickupObject | Pick up the remote that is on the blue chair |
|   | ToggleObject | Turn on the lamp with the remote in hand. |
| 3 | PickupObject | Pick up the laptop on the right after closing it. |
|   | ToggleObject | Turn on the floor lamp while carrying the laptop. |
| 4 | PickupObject | Pick the phone up from the desk. |
|   | ToggleObject | Turn the lamp on while holding the phone. |
| 5 | PickupObject | Grab the tissue paper from the dresser. |
|   | ToggleObject | Carry the tissue as you turn on the lamp. |
| 6 | PickupObject | Pick up the remote from the middle of the dresser, directly behind the tissues. |
|   | ToggleObject | Hold the remote and turn on the lamp. |
| 7 | PickupObject | Pick up a pillow from the chair |
|   | PutObject | Put the pillow on the couch |
| 8 | PickupObject | Pick up the statue on the top shelf. |
|   | ToggleObject | Turn on the lamp while holding the statue. |
| 9 | PickupObject | Pick up a statue from the dresser |
|   | ToggleObject | Turn on the floor lamp with the statue in hand |
| 10 | PickupObject | Pick up the left pillow on the chair |
|   | PutObject | Put the pillow on the sofa right of the newspaper |

Table 5: Tasks from Livingroom environment.

| Task No. | Sub-task Type | Instruction |
|---|---|---|
| 1 | PickupObject | Pick up the bar of soap on the back of the toilet. |
|   | PutObject | Place the soap in the trash can. |
| 2 | PickupObject | Pick up bar of soap |
|   | CleanObject | Put soap in sink, turn water on, turn water off, remove soap from sink |
| 3 | PickupObject | Pick up the cloth from the counter. |
|   | CleanObject | Put the cloth in the sink and turn the water on and then off and pick the cloth up from the sink. |
| 4 | PickupObject | Pick the soap up from the back of the toilet. |
|   | CleanObject | Put the soap in the sink and turn the water on and then off and pick up the soap again. |
| 5 | PickupObject | Pick the cloth up from the counter. |
|   | CleanObject | Put the cloth in the sink and turn the water on and then off and take the cloth out of the sink. |
| 6 | PickupObject | Pick up the bar of soap. |
|   | CleanObject | Put the bar of soap in the sink, turn the water on and then off and then pick up the bar of soap. |
| 7 | PickupObject | Pick up the bar of soap on the back of the toilet. |
|   | PutObject | Open the cabinet, put the bar of soap inside, and close the cabinet. |
| 8 | PickupObject | Grab a bar of soap off of the counter |
|   | PutObject | Put the soap in the trash can |
| 9 | PickupObject | Pick up the soap on the counter |
|   | PutObject | Open the cabinet and put in the soap then close the cabinet |
| 10 | PickupObject | Pick up toilet roll from off the toilet |
|   | PutObject | Open sink cabinet and place roll inside before closing the door |

Table 6: Tasks from Bathroom environment.

