# OpenReview forum: "VISION-LANGUAGE MODELS AS TRAINERS FOR INSTRUCTION-FOLLOWING AGENTS"
_ICLR.cc/2025/Conference — ICLR 2025 Conference Withdrawn Submission_

### Official Review · Reviewer_F1WB · 2024-11-03

**Soundness:** 3
**Presentation:** 1
**Contribution:** 2
**Rating:** 3
**Confidence:** 4

**Summary:**

Reward functions are burdensome for humans to design, especially when scaling up to complex environments and tasks. However, VLMs forgo the need for manual specification, allowing agents to learn diverse, language-specified tasks. Under this framework, V-TIFA uses VLMs for instruction-following agents in embodied tasks. V-TIFA shows that their way of extracting rewards from VLMs outperforms existing methods and enables competitive performance with ground-truth reward.

**Strengths:**

LC-RL with VLMs specifically for embodied navigation tasks appears to be a novel contribution. The prompting strategy, which includes extracting visual and textual prompts to further prompt a VLM, is interesting. Figure 4 is compelling, and Figure 5 shows interesting results on the precision of VLMs. The investigation on different types of VLMs also draws conclusions on the effectiveness of different pretrained models.

**Weaknesses:**

This paper is not very clear to follow. For instance, in the second paragraph of the introduction, the authors introduce RLHF, which feels only tangentially relevant to the actual method, which uses IQL (an offline RL adapted for online RL use, which should be further clarified in line 365, since readers will naturally assume IQL is used for offline RL). I don’t believe V-TIFA’s approach to be analogous to RLHF, except in some ablations, but the authors include many discussions of RLHF which are distracting and irrelevant. For instance, typical LLM RLHF focuses on comparison-based data from human annotators, which then learn a reward model, whereas this paper prompts VLMs for rewards, taking a more similar approach to robotics reward-learning papers such as Cui et al, Adeniji et al, Du et al, Shao et al, and Yang et al. In Section 6, the authors do provide an ablation for comparative feedback, but given how much the paper has focused on RLHF, the unpromising results here feel like a disappointing end to the problem setup.

In addition, the environment consists of primitive actions and a maximum of 50 environment steps. This isn’t inherently an issue, as using VLMs for embodied, higher-level tasks instead of low-level control is still interesting, but it does make the problem statement easier.

Visualizations of rewards or trajectories could be helpful in understanding where the VLMs make mistakes in generating ratings, and where they excel.

**Questions:**

1. Have the authors considered using VLMs for dense rewards? It appears that VLMs are being used for sparse rewards, like the ground-truth reward, since VLMs are used to obtain a final rating (line 256).
2. In line 243, don’t existing works directly incorporate VLMs into the training loop, if not for pretraining, but for finetuning? For example, Yang et al and Adeniji et al are two such works that use the reward model in the loop.
3. For Baselines, CLIP and R3M reward appear to only take in the initial and final frame of the trajectory. Is this not unfair, since V-TIFA can process the entire trajectory? R3M and CLIP have been applied as dense rewards, so shouldn’t the baselines likewise correspond to that?  For example, one could compute the CLIP reward across subtrajectories and take the average to receive a final rating. This would allow for more visual input than just the start and end frame of the trajectory.
5. Could the authors clarify language-assisted RL in line 33?

Adeniji, Ademi, et al. "Language reward modulation for pretraining reinforcement learning." arXiv preprint arXiv:2308.12270 (2023).

Cui, Yuchen, et al. "Can foundation models perform zero-shot task specification for robot manipulation?." Learning for dynamics and control conference. PMLR, 2022.

Du, Yuqing, et al. "Vision-language models as success detectors." arXiv preprint arXiv:2303.07280 (2023).

Shao, Lin, et al. "Concept2robot: Learning manipulation concepts from instructions and human demonstrations." The International Journal of Robotics Research 40.12-14 (2021): 1419-1434.

Yang, Jingyun, et al. "Robot fine-tuning made easy: Pre-training rewards and policies for autonomous real-world reinforcement learning." 2024 IEEE International Conference on Robotics and Automation (ICRA). IEEE, 2024.

---

### Official Review · Reviewer_JHv3 · 2024-11-04

**Soundness:** 2
**Presentation:** 3
**Contribution:** 2
**Rating:** 3
**Confidence:** 4

**Summary:**

The manuscript introduces the Vision-Language Models as Trainers for Instruction-Following Agents (V- TIFA) framework, which proposes to use foundation models for providing feedback in an Reinforcement Learning from AI Feedback (RLAIF) paradigm, for training agents to perform ALFRED tasks.

**Strengths:**

I like the general concepts that the manuscript proposes, including the use of VLMs for task (self-)supervision.

The paper is also well-written.

**Weaknesses:**

Abstract, Intro — The manuscript states, “Developing agents that can understand and follow language instructions is critical for effective and reliable human-AI collaboration” (L11-12) and “Recent approaches train these agents using reinforcement learning with infrequent environment rewards, placing a significant burden on environment designers to create language-conditioned reward functions.” (L12-15) Why would developing language-conditioned reward be the responsibility of the environment designers? The sparse rewards found in instruction-following tasks resembles quite closely the task-execution setting that humans are faced with naturally. Artificially providing agents with dense language-conditioned *extrinsic* rewards, from the environment, for ‘free’, would not be helpful for pursuing ‘effective and reliable human-AI collaboration.’

Abstract, Intro, Related Work — Many approaches attempt to use large-capacity models to generate rewards for training policies under an RL objective. Here are but a few: https://arxiv.org/pdf/2402.16181, https://arxiv.org/pdf/2402.04210, https://arxiv.org/pdf/2405.10292, https://arxiv.org/pdf/2409.20568, https://openreview.net/pdf?id=uydQ2W41KO. The manuscript should consider defining baselines based on these.

Abstract, Intro, Related Work — This manuscript is trying to propose ‘RL with artificial feedback’ — see “RLAIF vs. RLHF: Scaling Reinforcement Learning from Human Feedback with AI Feedback” | https://openreview.net/pdf?id=uydQ2W41KO

Section 1, Section 3 — The manuscript should make a distinction between extrinsic and intrinsic rewards, in instruction-following tasks; furthermore, the manuscript should be clearer about where the responsibility for good reward design lies: for extrinsic (environmental) rewards, the responsibility rests with the environment creators; these rewards are sparse for the purpose of realism (see also my previous comment(s)). The responsibility for intrinsic rewards rests with the agent/algorithm/methodology developers. The manuscript constantly conflates these two, inappropriately.

Section 4, Section 5 — I worry that the environment used by the manuscript for assessing the effectiveness of the approach is not sufficiently complex for serving as a good evaluation benchmark. The aforementioned approaches show good performance in their benchmarks as well. For example, ALFRED (the environment that this manuscript uses) is quite photo-*synthetic*, the tasks are short-horizon single-room tasks, and the benchmark has the luxury of including tasks with well-articulated implicit pre- and post-conditions and sub-goal segmentation. Any results on more complex, more photo-realistic, long-horizon, and/or less curated environments would be much more informative.

Section 5 — Experiments are insufficient. The choice of baselines illustrate comparisons in terms of the embedding spaces used for encoding the multimodal context, the but the evaluation does not provide comparisons with the prior art in terms of the fundamental methodological approach.

Section 5 — All of the tasks are short-horizon tasks. To illustrate the relevance and generality of the approach to more complex settings, experiments on long-horizon settings are needed (e.g., multi-room, multiple semantically dependent tasks in the same room, or at least multiple semantically independent tasks in the same room).

**Questions:**

No additional questions; please see above.

---

### Official Review · Reviewer_HXiR · 2024-11-05

**Soundness:** 2
**Presentation:** 3
**Contribution:** 2
**Rating:** 3
**Confidence:** 4

**Summary:**

This paper introduces a method that utilizes a large VLM as the reward function for RL training. It collects trajectories in advance, uses VLM to summarize and output a score that represents whether the trajectory successfully completes the given task. In ALFRED environment, V-TIFA demonstrates its effectiveness compared to baselines.

**Strengths:**

- Originality: V-TIFA innovatively proposes using a large VLM as a reward model for RL within a household simulation environment.
- Quality: Comprehensive experiments conducted in the ALFRED environment highlight V-TIFA's effectiveness, supported by an in-depth performance analysis.

**Weaknesses:**

- Originality: The originality of the proposed method is limited. 1. There has already been significant prior work using VLMs as reward functions for RL, such as MineDojo (Fan et al., 2022), which successfully completes a variety of tasks in the more complex, open-ended environment of Minecraft. 2. Using large VLMs to assess task completion has been widely applied in AI agent research, commonly referred to as success detection.

- Quality: The experiments in this paper are insufficient. 1. Experiments are conducted in only one environment, lacking results across multiple environments, which does not adequately demonstrate the method's effectiveness or broad applicability. 2. The baselines are unfair: V-TIFA is provided with a complete trajectory, while the baselines receive only the first and last frames, which is both simplistic and unfair. In MineDoJo (Fan et al., 2022), MineCLIP processes a continuous trajectory window, producing a dense reward. Even if the goal is to maintain a sparse reward setup, it would be fairer to select the highest score from CLIP or another model over the trajectory.

- Efficiency: V-TIFA is inefficient, as it only provides sparse rewards based on whether the trajectory completes the task. Due to RL’s inherent difficulty in handling sparse rewards, this approach is inefficient. Furthermore, using sparse rewards from the environment as an upper bound is not ideal. In MineDoJo (Fan et al., 2022), MineCLIP provides dense rewards to aid the agent in language-conditioned tasks that are challenging to complete using only sparse rewards from the environment. The primary reason for introducing LLMs or LVLMs into decision-making tasks is to improve sample efficiency and generalization in RL. However, here, the use of large VLMs does not surpass the sparse rewards provided by the environment, which limits the practical value of this approach.


References:

Fan, L., et al. (2022). Minedojo: Building open-ended embodied agents with internet-scale knowledge[J]. Advances in Neural Information Processing Systems, 2022, 35: 18343-18362.

**Questions:**

- Is there a randomized starting position for each scene? This is not clearly specified. In ALFRED, task difficulty primarily lies in locating the target object. If the starting position in a given environment is fixed, the agent could simply learn and memorize the static locations of objects, resulting in a policy that lacks generalization.
- How do the baselines perform when using dense rewards? Even within a shared sparse reward setting, selecting the highest score from CLIP, RoboCLIP, or R3M within a trajectory as the reward would offer a fairer comparison.

---

### Official Review · Reviewer_8szX · 2024-11-07

**Soundness:** 2
**Presentation:** 3
**Contribution:** 2
**Rating:** 5
**Confidence:** 4

**Summary:**

This paper proposes a simple recipe to provide reward supervision for training language-instruction following agents: score entire trajectories with off-the-shelf VLMs. With the last state of each trajectory assigned a numeric reward, an off-the-shelf reinforcement learning algorithm (the authors use IQL) can be used to learn a policy. The particular manner in which the VLM is tasked with assigning rewards is critical to the performance of the method. Image states in the trajectory are annotated with timestep indices, and the full sequence of images is fed into the context of the VLM. The VLM is queried twice, first to provide a textual summary of the input trajectory and second to assign a numeric reward based on the summary. In experiments conducted on the ALFRED simulator, training policies with RL using the proposed VLM reward generation framework leads to better performing policies than with baseline zero-shot VLM reward generation approaches.

**Strengths:**

The main strength of the proposed idea is its simplicity. As VLMs are trained to faithfully follow user instructions, directly prompting these models as is done in the paper is a sensible way to extract reward signal from the model. The manner in which the VLMs (e.g., Gemini) are prompted is also sensible and plays to the strengths of these VLMs, such as their long-context modeling capability and video/multi-image pre-training.

**Weaknesses:**

There are two primary weaknesses that I see in the paper: (1) the lack of discussion around potential failures of VLMs to perform zero-shot reward specification, and how these failures could be addressed, and (2) the lack of real-robot results.

(1) As far as the reviewer is aware it is not discussed in the paper how the proposed algorithm will work when the task being learnt is of sufficient complexity that off-the-shelf VLMs can no longer reliably assign correct rewards. The authors motivate the use of frozen VLMs for reward generation with the claim that it eliminates human effort spent towards collecting demonstrations or defining reward functions. However this is only true when the off-the-shelf VLM can indeed reliably specify rewards. If the VLMs' performance is sufficiently low to require SFT/RL finetuning on task-specific data to improve accuracy, then human effort cannot be avoided. Indeed this seems to explain the low performance of the baselines the authors evaluate against; the baselines do poorly because their frozen VLMs do not assign good rewards.

(2) Weakness (1) by itself is not a significant issue for me as there is much recent work that attempts to leverage frozen VLMs for control [1, 2, 3, 4, 5]. However like these works I would like to see real robot results, since these works have set the inclusion of real-robot results as a precedent.

[1] Nasiriany, Soroush, et al. "Pivot: Iterative visual prompting elicits actionable knowledge for vlms." arXiv preprint arXiv:2402.07872 (2024).
[2] Liu, Fangchen, et al. "Moka: Open-vocabulary robotic manipulation through mark-based visual prompting." First Workshop on Vision-Language Models for Navigation and Manipulation at ICRA 2024. 2024.
[3] Shah, Dhruv, Błażej Osiński, and Sergey Levine. "Lm-nav: Robotic navigation with large pre-trained models of language, vision, and action." Conference on robot learning. PMLR, 2023.
[4] Wang, Zidan, Rui Shen, and Bradly Stadie. "Solving Robotics Problems in Zero-Shot with Vision-Language Models." arXiv preprint arXiv:2407.19094 (2024).
[5] Hu, Yingdong, et al. "Look before you leap: Unveiling the power of gpt-4v in robotic vision-language planning." arXiv preprint arXiv:2311.17842 (2023).

**Questions:**

1. How can the proposed approach handle scenarios where the reward specification problem is beyond the capabilities of current off-the-shelf VLMs?
2. Can the authors demonstrate that their proposed approach can enable learning on a real robot? This might look something like collecting some real robot trajectories, labeling rewards with an off-the-shelf VLM, and training policies with IQL like is done in the paper currently except purely offline.

---

### Note · Authors · 2024-12-01

I have read and agree with the venue's withdrawal policy on behalf of myself and my co-authors.